



# Risk assessment and management for an extreme accident at a waste slag site

Shuang Liu[1], Bo Chai[1], Feng Luo[2], Lili Xiao[3]

[1] School of Environmental Studies, China University of Geosciences, Wuhan, Hubei 430074, People's Republic of China
[2] China Railway First Survey & Design Institute Group Co.,LTD., Xi'an, Shannxi 710075, People's Republic of China
[3] School of highway, Chang'an University, Xi'an, Shannxi 710064, People's Republic of China

*Correspondence to*: Bo Chai (chai1998@126.com)

**Abstract.** Waste slag failure is a disaster that affects both the environment and people. In China there are many waste slags after the engineering project such as railway and road. Although the disposal of the waste slags obeys the regulations the government have established, some accident still happened. Therefore, we need to do inverse analysis to check the failure extend and impact area of each slope. The probability of failure of a waste slag site involves many unpredictable factors that are hard to calculate. Therefore, we propose a risk analysis and management scheme for extreme accidents that assumes failure will arise at extreme conditions, and emphasize risk management in the design and monitoring of the slag site. In this scheme, we use Tsunami–Square Method to simulate the flow of tailings sand to get the intensity parameters (flow path and thickness) to create hazard zones based on dam failure. The risks to buildings and people were analysed according to the vulnerability of the buildings in the flow path. A "risk sharing community" risk management mode is presented using the idea of Canadian Whitehorse Mining Initiative, which sparkplug multi-stakeholder representatives to participate in risk management. Following the As Low As Reasonably Possible principle, the risk management scheme divides areas at risk into five zones in the F–N Curve. These zones have different mitigation measures for risk from tailings ponds and other waste slag sites. The scheme is effective for determining design safety factors, implementing reinforcements, and monitoring the waste slag site, and encouraging multiparty participation in risk management.

## 1 Introduction

A lot of high-risk waste slag is produced in the process of mine exploitation, road engineering, and urban construction. According to the statistics of the China Geological Survey (2009), only mining project has produced 17.6 billion tons of solid waste annually, forming thousands of large scale slags (tailings and dumps). Recent accidents of slag dump failures have led to significant economic losses and casualties(Sutherland et al., 2000;Foster et al., 2000;Madrid et al., 2006). In April 2005, a dam break at a phosphate mine near Bangs Lake in Jackson County, Mississippi, USA resulted in the collapse of a



phosphogypsum stack, causing 164,000 cubic meters of liquid acid to gush into nearby marshland, killing the plants. In November 2015, a tailings dam at a large iron and manganese mine owned by the Samarco Corporation collapsed in Brazil, generating a massive wave of toxic mud that spread down the Doce River, killing 19 people and affecting biodiversity across hundreds of kilometers of river, riparian land, and Atlantic coast(Garcia et al., 2017).

In China, the landslides and debris flow disasters in the waste slags and the dumping ground have brought serious consequences. On September 1962, Yunnan Tin Company Fire Valley tailings dam accident occurred, killing 263 people. On May 1970, Yanjinggou Iron debris in Sichuan killed 104 people. A small gold deposit in Qinling Mountains, Shaanxi in July 1994 in Henan at the junction of the debris flow, causing hundreds of casualties. On 8 September, 2008, a tailings dam failure at an iron ore mine in Xiangfen city, Shanxi Province, China, discharged 268,000 m$^3$ of tailings sand, forming a debris flow that

killed 277 people. The reason given for the dam failure was its incapacitated drainage system, which led to extensive liquefaction of the sand. On 20 December, 2015, a large spoil ground failed and formed a debris flow at the Guangming New District in Shenzhen, Guangdong, China. According to the analysis (Yin et al., 2016), the apparent friction angle of the spoil was about 6°, which is much lower than the test result of 30° (Xu et al., 2016). The failure of mine tailings dams and spoil grounds is a common problem that causes casualties and ecological destruction.

Every country has its own design standards and safety management measures for waste slag sites such as tailings ponds. In China, the "*Safety Technical Regulations for Tailing Pond*" (*STRTP*) clearly defines the technical requirements for their construction, operation, and closure to reuse, including monitoring, assessment, and safety factors. The International Commission on Large Dams summarized the causes of 185 major tailings dam failures in the USA, using the database established by United States Commission on Large Dams, and determined the design standard for tailings ponds. Europe

established a similar database to guide the design and construction of tailing dams (Rico et al., 2008). However, many tailings dam accidents still occur despite the strict technical standards (Table 1). According to statistics of 3500 tailings worldwide, Lemphers has revealed that 2-5 tailings dam failures every year, and the probability of dam failure is 10 times more than that of reservoir dam failure(Zhang, 2013).

Therefore, we must think about the following questions: is the safety factor of tailings ponds low in the technical standards?

Are there many factors, artificial or natural, affecting the stability of tailings pond but not being considered in their design? What measures, such as reinforcement or monitoring, are most effective at reducing the risk of tailings pond failure? Landslide susceptibility and hazard zoning have been developed since the 1970's(Brabb et al., 1972;Kienholz, 1978;Nilsen, 1979;Glade and Crozier, 1996;Dai et al., 2002). Landslide hazard and risk assessment is a wide field(Einstein, 1988;Guzzetti, 2000;Crozier and Glade, 2005;Claudio et al., 2007;Yin et al., 2007;Gibson et al., 2013). In 1994, Fell presented definitions of

risk and hazard with acceptable risk relating to other risks accepted by the community(Fell, 1994).In addition, land use planning(Fell et al., 2008) and different scales(Cascini, 2008;Van Westen et al., 2008) are important considerable factors in landslide susceptibility, hazard and risk zoning. Quantitative Assessment of Risk(QRA) is the best way to deal with estimating landslide hazard(Hungr, 2016) and the key components of QRA for landslide hazards, which allows scientists and engineers to quantify risk in an objective and reproducible manner and to compare the results from one location (site, region, etc.) with

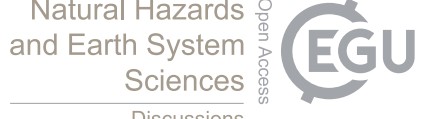



those from another(Corominas et al., 2013).Hungr also emphasized the workflow of QRA of landslide hazard and risk assessment should be conducted by geoscientists or geoengineers and others (owners or planners to determine the future development, structural engineers to help determine vulnerability of buildings and to design remedial structures). Although various methods are available for quantitative landslide risk analyses, applications are still rare and mostly dependent on the

occurrence of disasters(Bell and Glade, 2004;Yin et al., 2004). In addition, the difficult in determining the failure probability and lack of detail and quantify data of each program(Fell, 2016), multi-criteria decision analysis is used in many fields of risk management(Sterlacchini et al., 2007;Zamarrón-Mieza et al., 2017).

Compared with natural slope or strictly designed filled slope, the internal structure of slag (dump) is complex and has significant time-varying characteristics, which is difficult to accurately calculate its stability. The reason given for the failure

of the tailings pond in Xiangfen and the spoil ground in Shenzhen was that the fine particles in slag block the drainage system, reducing the strength at the bottom of slag pond. In addition, improper management during construction and operation are also the reasons for some tailings ponds accidents. However, these factors are usually difficult to predict at the design stage. If the design of a tailings pond improves its safety factor or increases some reinforcement structure for these uncertain factors, it would cost more and could not be accepted by the mine investor. For example, according to the ratio of the tested friction

angle (30°) and apparent friction angle 6° of spoil, the spoil ground in Shenzhen could not use a safety factor of > 3 to analyse its stability and design. The quantification of risk assessment and management of slag sites need to consider the probability of failure, intensity (runout distance, impact, and accumulating thickness), vulnerability of different elements, and risk analysis, assessment, and mitigation measures (Cardona, 2011). The analysis of the probability of failure, which is the basis of risk assessment and management, is difficult to determine because of these uncertain natural and human factors.

Moreover, after the spoil ground failed at the Guangming New District in Shenzhen, relevant departments worried about the safety problem of numerous spoil sites that were produced by mining, road building, and urban construction in China. These spoil grounds need risk post-assessment to decide if used safety standard can satisfy the present situation. These spoil grounds were piled up many year age and lacked geological data for failure analysis.

Therefore, we try to reduce the analysis of the probability of failure but improve risk management for lessening the risk of

accidents at waste slag sites. This paper discusses how to determine risk management by assessing the risk of an extreme accident, for example the failure of a tailings dam. Using the risk of an extreme accident to guide risk mitigation measures, a "risk sharing community" management mode is proposed, which suggests that miners, technical personnel, management, and locals should participate in risk management. This mode is consistent with the Whitehorse Mining Initiative in the environmental management of mines(Bowman and Baker, 1998), which tried to achieve risk mitigation through multi-

stakeholder consultations.

This study takes a copper mine in Daye mining, southeast Hubei, China as example, analysing the risk of an extreme accident and management schemes.



### 2 Study area and risk analysis of extreme tailings dam failures accidents

### 2.1 Study area

The study area is located in Chengui Town, which is in the center of Daye city, Hubei Province, China. The mine was built in 1984 and has an area of 3 km² (Fig. 1) including the exploration area and tailings pond. The minimum distance between the

tailings dam and residential areas is less than 50 m, which is a very high-risk situation.

The tailings pond was built in 1985 with a designed service life of 50 years. After being increased several times, the tailings dam is 82 m, with a capacity of $13.12 \times 10^6$ m³. At present, there are annual tailings emissions of approximately $9.0 \times 10^5$ m³. The main minerals include pyrite, hematite, chalcopyrite, and limonite (Liu et al., 2009).

### 2.2 The process of risk analysis

Tailings ponds have significant uncertainty, and dam failure accidents occur frequently. The probability of failure involves many unpredictable factors from the natural and human environments, and is hard to calculate. We modified the general risk analysis process to reduce analysis of the probability of tailings dam failures, but assumed that extreme accidents could occur. The process of assessing the risk of extreme accidents is shown in Fig. 2. In addition, it is suited to risk post-assessment of tailing ponds or others waste slag sites.

Step 1- Intensity and hazard analysis of tailings dam failures ($H$): using the statistics data for dam failures confirms damage scale of dam. Then, the experience model, indoor experiments, and numerical simulation is used to analyze the motion of tailings sand and to determine parameters of intensity, such as velocity, mud depth, and reach. In this study, we use indoor experiments and numerical simulation to analyse these parameters. In addition, the hazards of tailings dam failures are analyzed according to the hazard classification of the debris flow (Varnes, 1978;Cruden and Varnes, 1996;Fiebiger, 1997;Hungr et al.,

20   2001).

Step 2-Vulnerability analysis of elements at risk ($V$): people, buildings, roads, and land in the path of moving tailings sand are the main elements at risk. The attributes and classification standards of the vulnerability assessment of each element follow that in the risk of debris flow. In this study, the number of buildings and residents was investigated in the field, and the owners of the buildings are defined as the main participant in the risk management.

Step 3-Risk analysis of tailings dam failure: according to Eq. (1), risk is determined by the hazards of tailings pond failure and the vulnerability of buildings. Risk level of each building is determined by general international risk classification standards:

$$R=H \times V \tag{1}$$

where $R$ is risk, $H$ is hazard of tailings pond failure, and $V$ is vulnerability of element in risk.

Step 4 to Step 6- Risk assessment and management of tailings pond failures: the general purpose of risk assessment is to

determine acceptable risk (or risk tolerance). Risk assessment is based on casualties and property losses caused by a disaster that are determined as acceptable by economists, sociologists, mine owners, and government departments at Step 4.





At Step 5, risk management consists of risk decision-making, control, and implementation. Risk decision-making comes after risk assessment to make designs responsive to risk, determine which designs present acceptable risk, avoiding risk, and transferring risk. Risk control refers to the specific measures to reduce risk, generally divided into three categories: implementing engineering measures to reduce hazards from tailings ponds; reducing the vulnerability of elements at risk, and reducing and dispersing the consequences of disasters. The research emphasis of this study is risk management. Based on the risk analysis of tailing pond, this paper will discuss how to determine the design safety factors of tailing dam to make them less hazardous, involve local residents to participate effectively in demonstrations of tailings pond operation, and monitor the operation period to disperse the consequences of disasters. The safety factor of tailing dam is key parameter to decide its reliability and failure probability at Step 6.

## 3  Results of risk analysis of tailings dam failure

### 3.1 Hazard assessment of tailings dam failure

#### 3.1.1  Parameters of moving tailings sand

According to the actual size of the tailings dam, the indoor experimental model was designed at a 1:750 ratio. The tailings sand was taken from the field, and the concentration of the experimental tailings pulp was 40%. We designed tailings dam failures based on different scopes, ground slopes, and ground roughness, according to the scope of failures in some accidents, and used a high-speed camera to record the process and final form of the tailings sand flow (Fig. 3).

The Tsunami–Square (T–S) method(Xiao et al., 2015), which is adequate for various types of flow and flow-like behavior, was used in the experimental indoor simulation. A comparative analysis of the lab experiment runoff and the simulated results determines the three basic motion parameters of the tailings sand, namely the basal friction coefficient ($\mu_b$), dynamic friction ($\mu_d$), and angle of repose ($\theta_r$). The method and its motion parameters were then used in the failure motion of actual tailings pond.

#### 3.1.2 Intensity parameters of tailings dam failures

Based on digital elevation model data from the study area, a 3D terrain model was established using the T–S method. Estimating the scope of a 1/2 and 1/4 breach in the tailings dam, we obtained the thickness and velocity of moving tailings sand, the main parameters of intensity in cases of dam failure (Fig. 4).

The movement of the tailings sand is related to the terrain in front of the dam. The sand flowed north, and stopped when it reached the hill northeast of the tailings pond. Increasing the scope of failure increased the run-out distance and intensity parameters of the tailings sand. The tailings are thick in the middle and thin at the edges. The maximum speed is mainly near the site of the dam burst and rapidly decreases with run-out distance.





### 3.1.3 Hazard zoning in tailings dam failure

The intensity classification scheme for debris flow(Fiebiger, 1997) uses mud depth, and its relationship to the maximum velocity, as the classification factors. In addition, using Austria and Switzerland as references, the hazard zones for tailings dam failures, based on the debris flow intensity criteria (Table 2), can be obtained from simulation results using the T–S method (Fig. 4).

### 3.2 Vulnerability and Risk assessment of buildings

According to vulnerability assessment indexes of landslide risk, building vulnerability is based mainly on their construction, maintenance, and service life(Uzielli et al., 2008). According to the actual situation, buildings are divided into four categories. The maintenance conditions are related to the deformation of buildings and the service life year is the ratio of completion life and design life. Using Eq. (1) calculates the risk from a 1/4 and 1/2 breach in the dam. The result of the risk analysis is shown in Fig. 5 and Table 3.

The economic losses use the number of buildings damaged, with one building costing 100,000 CNY that is determined by investigating and market analysis. In this case, the degree of damage to buildings is 0.7–1.0, 0.4–0.7, 0.1–0.4, and 0–0.1 in the high-, moderate to high, moderate, and low-risk areas, respectively. We can take the intermediate value as the probability of damage to buildings and determine this factor for different risk areas. 16.4 buildings that might be damaged under a 1/4 breach is, with economic losses of 1.64 million CNY; under a 1/2 breach the value of the former is 32.2, with economic losses of 3.22 million CNY. The temporal probability of impact on people is 0.6. The likelihood of loss of life is 0.5 in the high-risk area and 0.15 in the moderate to high risk area. The probability of casualties in the area is 3.75 under a 1/4 breach and 38.94 under a 1/2 breach.

### 4 Discussion: risk management schemes for extreme accidents

According to the Canadian Whitehorse Mining Initiative (WMI), mine closure plans need the public participation and should include mine disaster management. The WMI is a multi-stakeholder consultation process designed for mining in Canada. Representatives included those from the mining industry, senior government officials, labour unions, Aboriginal peoples, and the environmental community (Bowman and Baker, 1998). The goal was to move toward a socially, economically, and environmentally sustainable and prosperous mineral industry, underpinned by political and community consensus. For high-risk structures such as tailings ponds, determining the acceptable risk and risk management plans also needs multi-party participation. Risk management, from deciding acceptable risk levels to implementing mitigation measures, needs multi-stakeholder consultation. It could be a "risk sharing community," suggesting that miners, technical personnel, management departments, and residents should all participate in risk management.

(1) How should the acceptable risk be determined?





In landslide risk assessment, the F–N curve is used to determine the degree of acceptable risk. For risks from tailings ponds, F is the probability of failure of a tailings dam, which is difficult to determine because there are many unpredictable factors. N is the number of deaths or economic losses in the case of a dam failure; this can be determined based on the number of buildings in the path of moving tailings sand.

The F–N curve needs to accord with the As Low As Reasonably Practicable (ALARP) principle; if acceptable and unacceptable risk values are provided, a value between these will meet it. Different countries and regions have different risk tolerance criteria for disaster accidents. According to the principle of multi-party participation, we propose that the risk tolerance criteria should be given by experts. However, input should also be taken from local residents, governments, and the mineral company, because this factor is related to the economic capacity to implement safety measures at the dam, and for the people and buildings in the

hazard area.

The acceptable risk for people is obviously far less than that for buildings, and it is critical to determine this level for management purposes. According the consequences of the disaster, and policies and regulations in the area, the standards of acceptable risk are usually determined by expert analysis. However, waste slog sites are man-made and therefore more unpredictable than natural disasters. Residents need to be assured of the security at these sites, but this cannot be out of a range

suggested by experts. Therefore, we propose that the upper (unacceptable) limit could be determined by experts, but that suggestions from residents should be considered for the lower (broadly acceptable) level. The government and mineral companies can act as intermediaries in determining the lower boundary of the acceptable risk. The risk buffer zone, for people and buildings, should be discussed by the multiparty representatives participating in the risk assessment (Fig. 6a).

(2) How should risk mitigation measures be chosen?

The risk management scheme is divided into five zones on the F–N Curve. These zones use different measures to mitigate the risk from tailings ponds and other waste slag sites (Table 4).

(a) Zones Ⅰ and Ⅱ are areas of unacceptable risk, indicating that the tailings ponds are unsuitable in their current state. The lowest safety factor of the dam could be decided at the design stage, based on economic losses and fatalities in extreme accidents. If the safety factor subsists and unchangeable, some buildings should be moved out the risk area.

(b) Zone III is an area of intermediate risk, in which the risk to buildings range from broadly acceptable to unacceptable; these are areas where risk is tolerable and can be managed by appropriate measures to reduce it. Reducing the time people spend in these areas, reinforcing buildings, and monitoring dam deformation are effective measures to reduce risk here. Reinforcing sections of the tailings dams that can affect densely populated areas can also reduce risks to people and from economic loss. The costs of reinforcements to buildings or the dam could place some cost burden on mineral companies; therefore, some

economical reinforcement measures suggested by technical personnel could be adopted.

(c) In zone IV, economic losses are an acceptable risk and losses of life is a tolerable risk. The buildings and the dam do not need reinforcement. Reducing the time people spend in this area and monitoring dam deformation are the main measures to reduce risk to persons. Finally, Zone Ⅴ is an area where risks are acceptable in terms of economic losses and casualties, and mitigation is not needed.





In risk management of the tailings pond, when the breach changes from a 1/4 to a 1/2, the risk to people reduces greatly and the risk of economic loss changes little (Fig.6 b). Therefore, we need to strengthen tailings dams to control the breach to a 1/4 level in extreme circumstances. When the degree of reliability of dam stability is lower than 3.85, the building risk is unacceptable. Two measures could be chosen, i.e. strengthening whole dam or relocating the buildings. When its reliability is

between 3.85 to 4.00, dam need local strengthening. We should combine dam reinforcement (decrease the probability of failure) and measures taken in the area of influence to determine which is more cost-efficient. When the dam reliability is between 4.00 to 5.00, it is located in the Ⅲ and Ⅳ. The buildings reinforcements and dam monitoring is effective. Once monitoring equipment detects a threat from the dam, this area should be evacuated as soon as possible.

(3) How to do risk post-assessment of a waste slag site?

According to the *CHINA STATISTICAL YEARBOOK* in 2016, urban construction land area of China was 51584.1 km². The mileage of high-speed railway was 19838 km and highway was 123500 km. The abandoned mine development area was 7987.5 km². The surrounding environment and internal structure of accumulated tailings or other slags has changed over time. The risk assessment of existing slags is needed.

A key content of post risk assessment is to analyze whether the design safety factor of tailings dam and other supporting works

meets the requirements, and whether it is necessary to take corresponding risk mitigation measures. The safety factor used in the design stage often only considers the environmental uncertainty and slag parameter uncertainty. A large number of disaster events have been proved that the traditional design safety factor covered by uncertain factors is obviously insufficient. It also should consider the trend of risk change. The change of risk is mainly caused by changing in surrounding land use, trigger factors (rainfall, artificial loading, etc.) and internal structure (slag structure and retaining structure). (Fig. 7 and Fig.6a)

i) To consider the change of surrounding land use: when the land around the slag field is changed from wasteland to construction land. The risk would increase. The safety factor of the original design may not meet the requirement, and we need to reinforce the dam or other measures to improve the site safety factor.

ii) To consider the change of trigger factors: rainfall and other natural factors will not change significantly in a short time, but human activities tend to change greatly. For example, the storage area of Guangming New District continues to expand, and

the load of the dam increases continuously, making the possibility of instability of the dam increase.

iii) To consider the changes of slag structure: there are particle separation and structural zoning in the process of forming slags. The pressure and seepage will cause the long-term change of the particle structure. Due to the inhomogeneity of slag structure, the direction of slag structure change is different, and the variation modes of slag strength parameters are different. In the process of particle separation and compaction of newly accumulated slag, the dynamic and static pore water pressure change

is the key to induce slag heap structure instability. In some waste slags such as Shenzhen Guangming New District residue field are associated with poor drainage system. When the slag heap drainage system efficiency is reduced, the internal pore water pressure will have a great rise. The extreme form of excess pore water pressure in local makes the slag stability decrease. The change of the internal structure of the slag body is the key to the analysis of the probability of the instability of the slag heap.





iv) To consider the change of dam structure: the aging of supporting materials, the local damage and deformation and the increase of internal water content. If it exceeds the safety category of the original design, we need to recheck and reinforce the stability of dam.

According to the regulations *STRTP* of China, the safety degrees of tailing pond depends on flood control capacity and tailing
dam stability. This paper mainly discusses the risk of tailing dam instability and the management of the tailing dam,the flood control capacity is regarded as an influence factor of tailing dam stability. The regulations define the stability analysis of dam using Ordinary method and Bishop's Simplified method. In non-seismic region, the limit equilibrium equation of Ordinary method $Z$ is defined as:

$$Z = \sum_{i=1}^{n} [\frac{c_i l_i}{Fs} + (W_i \cos \alpha_i - U_i) \frac{\tan \varphi_i}{Fs} - (W_i \sin \alpha_i)] \qquad (2)$$

Where, $W_i$ is $i$ slice weight; $c_i$, $\varphi_i$, $l_i$, $\alpha_i$ and $U_i$ are cohesion, friction angle, area, dip angle and uplift pressure of slip surface on $i$ slice, $Fs$ is factor of safety; n is the number of all slices.

The factors $c$, $\varphi$, $U$ and $Fs$ are defines by the regulations *STRTP* and "*Code for investigation of geotechnical engineering*" , which are listed in Table 5. Based on test data, the design $c$ and $\varphi$ are determined by calculating the standard values on $\alpha = 0.05$ and the grade of tailing pond.

$$f_d = f_k / \gamma = \left[ f_m \pm \left( \frac{1.704}{\sqrt{n}} + \frac{4.678}{n^2} \right) \sigma_f \right] \Big/ \gamma \qquad (3)$$

Where: $f_d$ and $f_k$ are design value, standard value of shear strength respectively; $n$, $f_m$ and $\sigma_f$ are the number, mean value and standard deviation of testing shear strength; $\gamma$ is the correction factor.

In the design, the regulations are the basis of calculating stability of tailing dam. When the dam stability analysis considers DFF and corresponding factor of safety, the Eq.(2) satisfy limit equilibrium Z=0. However, it doesn't consider the failure
probability. According to previous studies, shear strength and rainfall are main uncertainty factors in the limit equilibrium equation. Therefore, the failure probability of tailing dam $P_f$ is expressed as:

$$P_f = P(Z < 0) \quad and \quad Z = g(c, \varphi, U, Fs) \qquad (4)$$

When the uplift pressure $U$ is determined by DFF, the upper and lower limits of the instability of the tailing dam could be
    written as:





$$P_{f\,\text{max}} = \left[\frac{1 - P(c \geq c_d/Fs) \cdot P(\tan \varphi \geq \tan \varphi_d/Fs)}{DFF}\right]$$

$$P_{f\,\text{min}} = \left[\frac{P(c < c_d/Fs) \cdot P(\tan \varphi < \tan \varphi_d/Fs)}{DFF}\right] \tag{5}$$

Where $c_d$ and $\varphi_d$ are the design values.

For the tailings pond in study area, it belongs to grade III with storage capacity $13.12 \times 10^6 \text{m}^3$ and dam height 82 m. The dam stability analysis is DFF 200-500 and corresponding factor of safety is 1.10. In the design stage, the mean and variance of the

shear strength of tailings sand are 0.287, 0.06 for $\tan \varphi$ and 10.50, 2.00 for $c$. The experimental number n=10 does not meet the statistical requirements of determining probability distribution. Previous studies suggest that $\tan \varphi$ and $c$ of geo-materials matches the normal or logarithmic normal distribution. According to Eq. (5) and Eq. (3), the probability of failure of tailings dam are calculated and shown in Table 6. Since the $\gamma$ value is not specified in the regulations, it is given a range value 1-1.6. The regulations *STRTP* just considers the scale of tailing pond for defining its grades and doesn't consider the

elements at risk. So, the correction factor $\gamma$ could be employed to adjust the factor of safety. The probability of failure of tailings dam with normal distribution are drawn in Fig. 6b. The upper and lower limits of $P_f$ run through the zone of ALAPR when the $\gamma$ is 1.6. In the working period of tailing pond, there is a phenomenon of fine particles aggregation in tailing sand, and the shear strength parameters tend to be lognormal distribution and lower expected values. If the expected values of shear strength is 95% of mean value, the upper and lower limits of $P_f$ run through the zone of ALAPR with $\gamma$ is 1.4(Table 6). So,

the $\gamma = 1.5$ is suitable correction factor for the design of tailing pond and long-term stability.

(4) How should monitoring and early warnings be implemented?

Monitoring is the main measure of tolerable risk in zones II, III, and IV. Monitoring and early warnings could ensure residents evacuate before dam failure occurs, and reduce the time people spend in a risky area. In addition, monitoring could help determine whether the dam is deformed or not, allowing the implementation of timely remedial work to prevent destabilization

accidents. We suggest that mineral companies employ local residents to participate in daily inspections, and technical personnel to monitor dam deformation. The locals could be better at finding tiny signs, indicative of dam stability, that could be effective for early warnings. However, an expert team is necessary to determine the need early warning and the corresponding emergency. The status of tailings dams need to be monitored every 24 hours, and if a danger is found, emergency disaster response mechanism should be started immediately.

The focus of the ordinary inspection of the local resident are:

i) Free board and minimum beach width on rainfall period;

ii) Whether the drainage system is effective and whether there are signs of deformation of the drainage well;

iii) Whether the dam has signs of deformation, piping, seepage, water turbidity and sliding;



iv) Whether there is rain erosion in the slope of the dam.

## 5 Conclusion

Residue fields, such as tailings and slag heaps, are artificial areas of high risk. Compared to natural disasters, accidents due to instability at these sites are more dependent on human factors, such as maintenance of drainage facilities and the quality of reinforcement engineering. Scientific management is the key to reducing the risk from slag heaps. The extreme event risk management model proposed in this study uses calculations the probability of failure, which can effectively guide the design of slag yard reinforcements and other risk mitigation projects, in risk management. This model also proposed the concept of a "risk sharing community," to encourage participation from mineral companies, residents, technicians, experts, and government departments involved in risk mitigation to eliminate the artificial influence of uncertainty. Risk management is an effective model in many ways, and this study uses a tailings dam breach event as an example to explain how. Practical application of this model should be combined with the actual characteristics of the slag field and further study of the hazards, vulnerability, reliability, and safety coefficient of the dam design index.

### Acknowledgement

The research presented in the paper was funded by the National Natural Science Foundation of China (Grant No. 41572256) and the Fundamental Research Funds for the Central Universities, China University of Geosciences (Wuhan) (Grant No. CUGL160408).

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





**Figures**

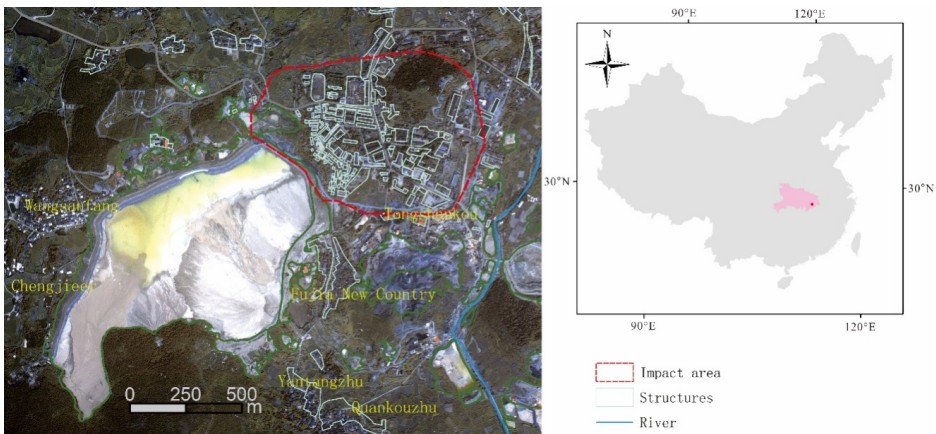

**Figure 1: General map of the mining area in Chengui Town**


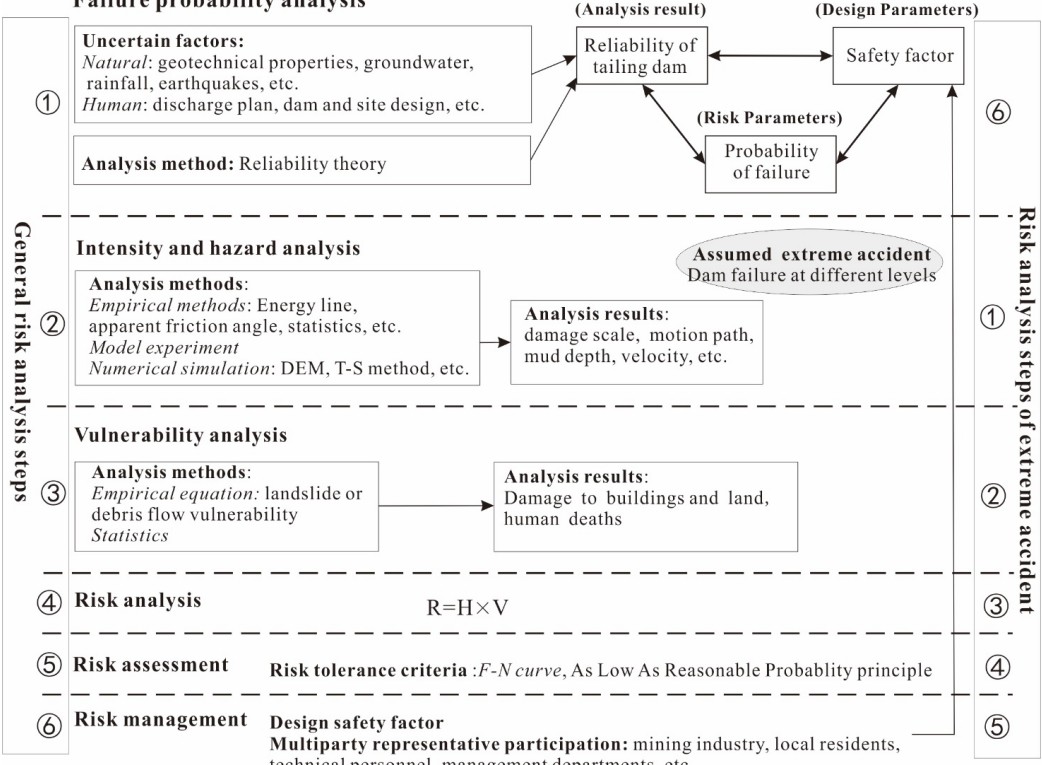

**Figure 2:  The risk assessment process**



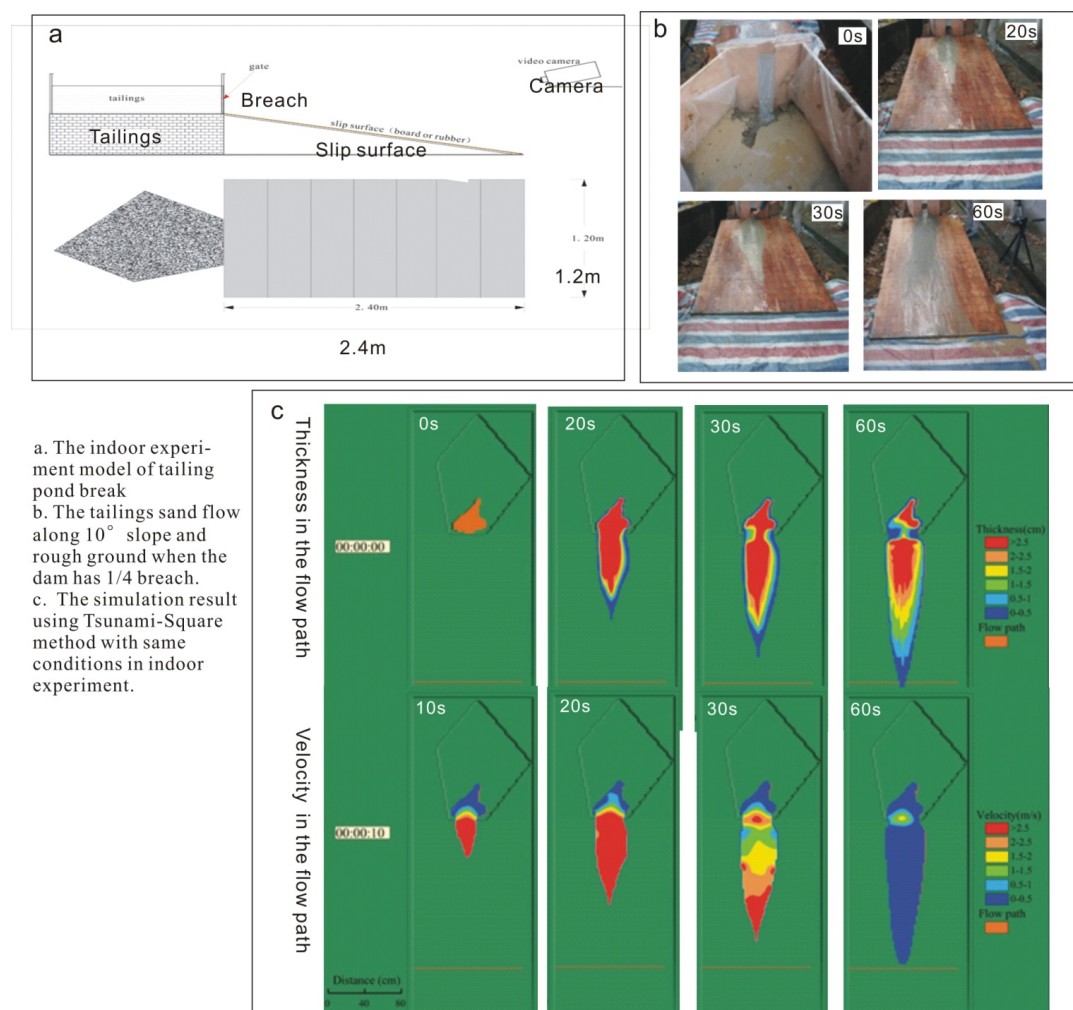

a. The indoor experiment model of tailing pond break
b. The tailings sand flow along 10° slope and rough ground when the dam has 1/4 breach.
c. The simulation result using Tsunami-Square method with same conditions in indoor experiment.

**Figure 3:    The experiment model and simulation results of T-S method**



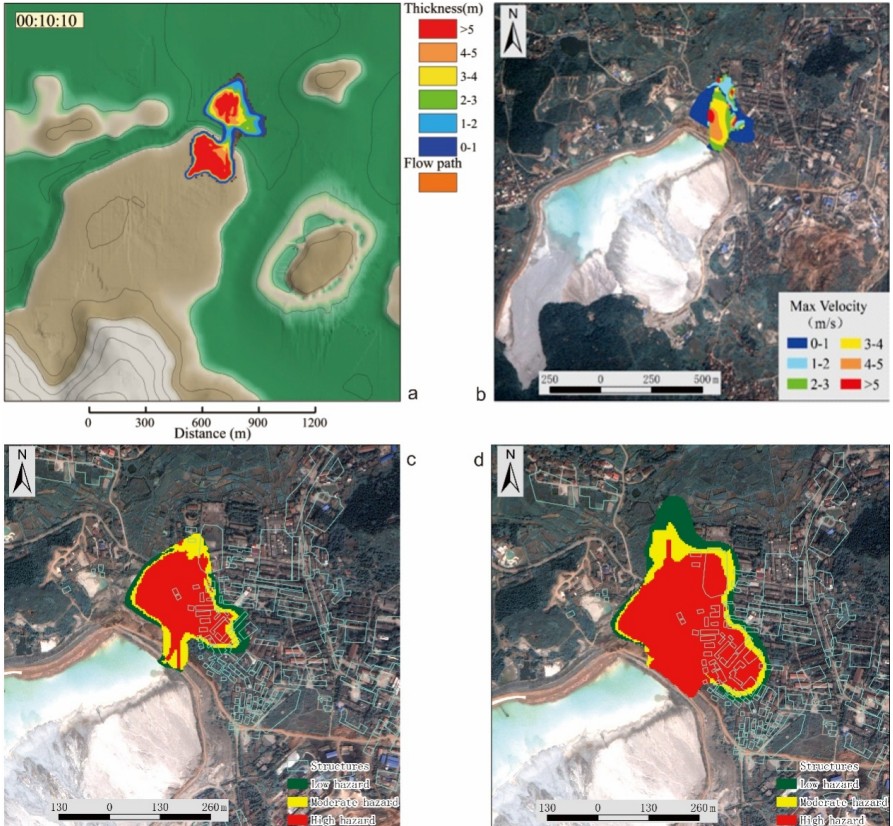

**Figure 4: Depth (a) and maximum velocity (b) of mud during a 1/4 dam breach accident; Hazard zones after 1/4 (c) and 1/2 (d) breach in a tailings dam**



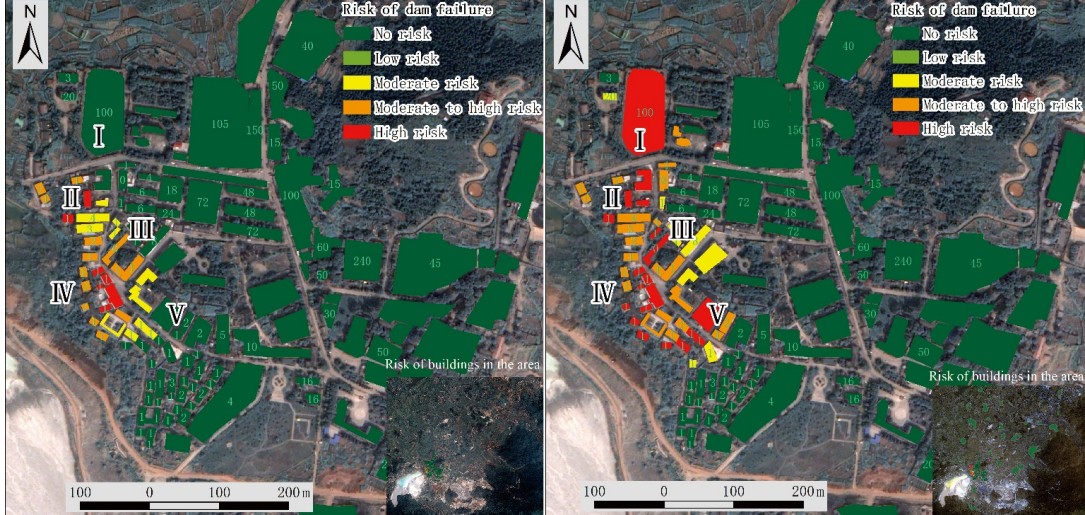

**Figure 5: Risk zones after a 1/4 and 1/2 breach at a tailings dam**



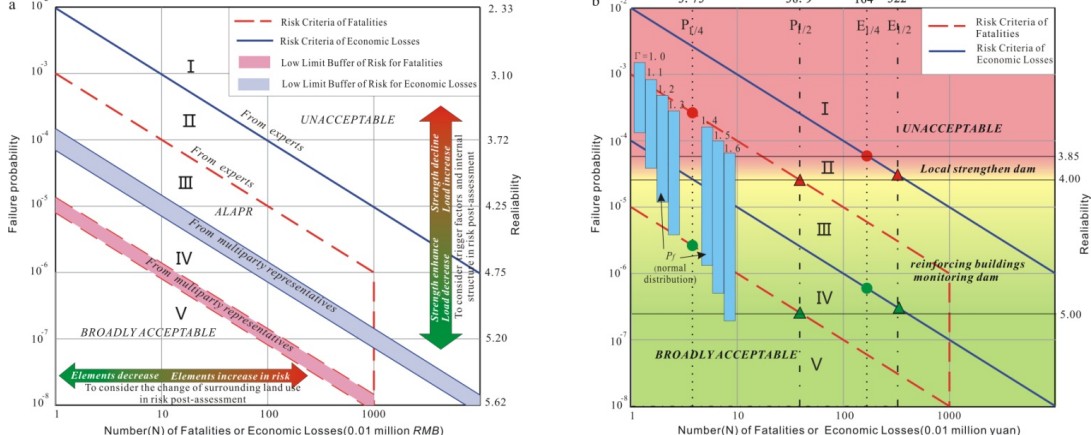

**Figure 6: (a) Dam break risk tolerance criteria and (b) Risk management and mitigation in the case: P1/2 and E1/2 are fatalities or economic losses under a 1/2 tailings dam breach, P1/4 and E1/4 are fatalities or economic losses under a 1/4 tailings dam breach. According to reliability or failure probability, four risk mitigation measures are determined, i.e. unacceptable, local strengthen dam, reinforcing buildings-monitoring dam and broadly acceptable. The upper and lower limits of failure probability with different correction factors are shown by blue columns when the shear strength parameters matches the normal distribution.**



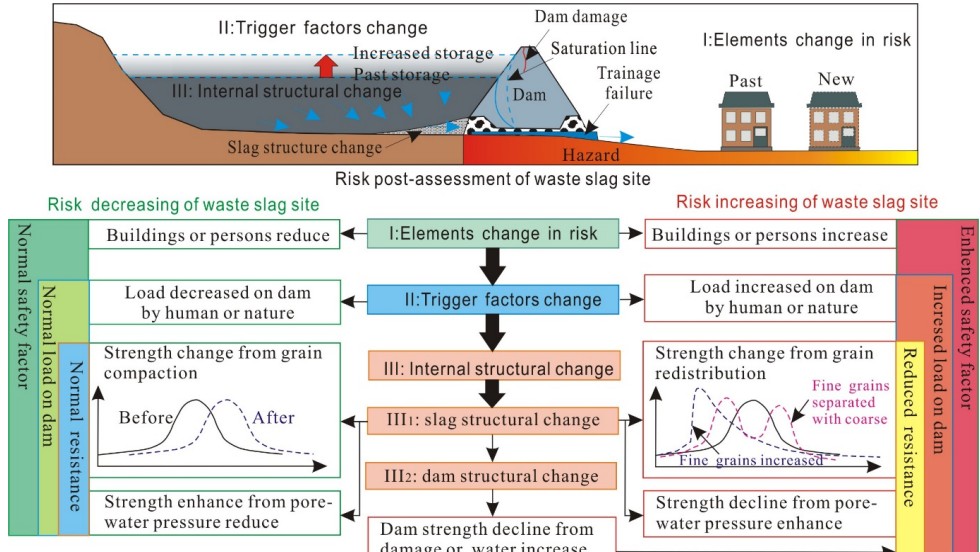

**Figure 7:** Flow chart for post-risk assessment of slags: the upper part of the section describes the distribution of slags risk factors, the lower figure shows two trends of slags risk factors change and risk change




## Tables

**Table 1 Slag dam break accidents**

| Location | Slag type | Date | Volume(m³) | Consequences | Reference |
|---|---|---|---|---|---|
| Shenzhen, China | Building slag | 12.20.2015 | $2.73 \times 10^6$ | Damaged 33 buildings, 77 fatalities | (Yin et al., 2016;Xu et al., 2016) |
| Tibet, China | Mine slag | 31.3.2013 | $0.20 \times 10^6$ | 83 fatalities | |
| Shanxi, China | Mine tailings | 9.8.2008 | $0.268 \times 10^6$ | 277 fatalities | (Yin, 2008) |
| Bandung, Indonesia | Waste material landfill | 2.21.2005 | $2.7 \times 10^6$ | Buried 71 houses, 143 fatalities | (Lavigne et al., 2014) |
| Quezon City, Philippines | Municipal soil waste | 7.10.2000 | $1.6 \times 10^4$ | More than 330 fatalities | (Merry et al., 2005;Jafari et al., 2013) |

**Table 2 Scheme for dividing the geological environment into hazard zones after a tailings dam failure**

| Hazard level | Mud depth (m) | Relationship | Mud depth and maximum velocity | Hazard assignment |
|---|---|---|---|---|
| High | H > 2.5 m | OR | VH ≥ 2.5 | 1 |
| Moderate | 0.5 ≤ H < 2.5 | AND | 0.5 ≤ VH < 2.5 | 0.7 |
| Low | 0.0 ≤ H < 2.5 | AND | VH < 0.5 | 0.4 |

**Table 3 Risk statistics of different breach forms in front of dam in dam break**

| Breach forms | 1/4 breach | 1/2 breach |
|---|---|---|
| | Number of structures (persons) affected at different risk levels | |
| High risk | 7 (5) | 21 (113) |
| Moderate to high risk | 14 (13) | 22 (25) |
| Moderate risk | 11 (12) | 9 (31) |
| Low risk | 0 (0) | 0 (0) |
| No risk | 57 (1,830) | 37 (1,691) |

**Table 4 Risk mitigation measures to be taken in risk zones**

| Risk zone | Mitigation measures |
|---|---|
| I | No people or structures should be in the area. Some monitoring instruments should be installed to determine whether or not the dam has failed. |
| II | Some structures could be built in this area, but people should not live or work in them for long periods. This area should be monitored, and an early warning system put in place, so that the people working here can evacuate in time. |
| III& IV | Some technology experts who have to work in this area could live here. Apart from monitoring and early warning projects, patrol personnel are indispensable here. |
| V | People can live and work in this area. The warning and daily inspections should monitor this area. |

**Table 5 Design parameters of tailing pond in the regulations *STRTP***



| Grades of T.P. | Grades' Criteria | | Design Flood Frequency (DFF) | $Fs$ of main structure | |
| --- | --- | --- | --- | --- | --- |
| | Storage capacity V($10^6$m³) | Dam height H(m) | | normal | Flood |
| I | Special Requirements | | 1000-2000 | 1.30 | 1.20 |
| II | V≥100 | H≥100 | 500-1000 | 1.25 | 1.15 |
| III | 10≤V＜100 | 60≤H＜100 | 200-500 | 1.20 | 1.10 |
| VI | 1≤V＜10 | 30≤H＜60 | 100-200 | 1.15 | 1.05 |
| V | V＜1 | H＜30 | 50-100 | | |

**Table 6  The upper and lower limits of failure probability of the tailings pond**

| PARM | Value | $\gamma$ | Normal | | Lognormal | |
| --- | --- | --- | --- | --- | --- | --- |
| | | | $P_{f\,max}$ | $P_{f\,min}$ | $P_{f\,max}$ | $P_{f\,min}$ |
| DFF | 200 | 1 | 1.49E-03 | 1.30E-04 | 1.78E-03 | 1.82E-04 |
| $Fs$ | 1.1 | 1.1 | 8.41E-04 | 3.85E-05 | 9.12E-04 | 3.96E-05 |
| Mean $\tan\varphi$ | 0.287 | 1.2 | 4.75E-04 | 1.17E-05 | 4.18E-04 | 7.03E-06 |
| Variance $\tan\varphi$ | 0.06 | 1.3 | 2.75E-04 | 3.79E-06 | 1.80E-04 | 1.09E-06 |
| Mean $c$ | 10.5 | 1.4 | 1.65E-04 | 1.32E-06 | 7.39E-05 | 1.49E-07 |
| Variance $c$ | 2.00 | 1.5 | 1.02E-04 | 4.91E-07 | 2.95E-05 | 1.82E-08 |
| | | 1.6 | 6.43E-05 | 1.93E-07 | 1.20E-05 | 2.20E-09 |