# Peer review of "Risk assessment and management for an extreme accident at a waste slag site"

_Natural Hazards and Earth System Sciences, 2018_

## Referee Comment (RC1) · Anonymous Referee #1 · 13 Jul 2018

The article deals with a relevant topic, risk management of waste slag repositories. More specifically, a procedure is presented to assess and manage the risk posed by tailing dams failures and a case study is presented. However, the paper does not properly highlight the innovative contribution of the proposal and its application to the case study is not clearly presented. Therefore, I suggest to thoroughly restructure the paper and to resubmit it after a major revision. Detailed comments follow.

MAIN COMMENTS

The article is difficult to follow for the reader, as its content alternates back and forth between sections describing the features of the proposed procedure and sections illustrating the case study. For instance, the test area is presented right after the Introduction (Section 2.1, Figure 1), yet the results related to the application of procedure appear

in sections 3.1.3 (Figure 4), 3.2 (Table3, Figure 5) and also in the Discussion section (Figure 6). Moreover, current section 4 "Discussion: risk management schemes for extreme accidents", include lengthy parts describing key elements of the procedure, including how to use risk acceptability criteria (zone of F-N space) and how to compute the stability of the tailing dams. These parts should be presented before. The article should be thoroughly restructured, so as to present the procedure first, then its application to the case study, then a discussion section clearly highlighting advantages and limitations of proposed contribution.

The procedure is presented as a "risk sharing community" risk management mode. Yet, the role played by the different stakeholders does not clearly emerge from the . More details are needed on this central aspect of the proposed procedure. A section (or subsection) should be devoted to this, when the procedure is presented, including all the text (and the Figure) currently placed at the beginning of the Discussion section (pages 6-7). Figure 6. Improve legibility of Figure. In the current version it is not possible to read most of the text. Retain only what is needed and explained (in the caption and in the text).

The originality of the overall procedure is not clear, as the key innovative elements proposed are not properly highlighted. For instance, the Authors refer to the Canadian Whitehorse Mining initiative (WMI). What are the elements from that initiative that are included in the proposed procedure? How does the proposed procedure differ (or adhere) from WMI? The STRTP regulations are also recalled a couple of times, for the risk zones in the F-N space and for the stability computations. Clearly define where and how they play an essential role in the procedure. Other methods proposed in the literature are also used: the Tsunami-Square method (Xiao et al., 2015), the intensity classification scheme for debris flow (Fiebiger, 1997), vulnerability assessment indexes (Uzielli et al., 2008). Some details on the main features of these methods must be provided and, once again, clearly define if they have an essential role in the procedure or if they are only used to apply the procedure to the case study.

[Figure]

Figure 2 should be restructured to better highlight the phases of the procedure described in the text. The current version is confusing because it start at the top with elements that are indicative of the last phase of the procedure (step 6). The same terms should be used in the text and in the Figure. For instance: the term "step", used in the current version of the text to describe the 6 phases of the procedure, does not appear at all in the Figure; the term "intensity analysis" only appear in the text.

The results of the procedure applied to the case study must be described following the different phases of the procedure, as they are presented in Figure 2. For instance, the subsections of this section could be six, one for each step of the procedure.

The role of the laboratory experiment must be better explained. Many more details on the characteristics of the performed experiment are needed. How does it reproduce the characteristics of the tailing pond of the case study? How many experiments were run? How were they different? The main experimental results (e.g. sections with height of deposited material) must be reported in graphs. A graphical comparison between the experimental test(s) and the numerical analysis with the Tsunami-Square method must also be reported, highlighting the role of the model input parameters and reporting the parameter values for the best-performing model simulation(s).

Introduction. It is too long and not well balanced in terms of topics addressed. For instance, the part reporting a detailed discussion on the failure of the "tailings pond in Xiangfen" and "the spoil ground in Shenzhen" is not appropriate. Also the reference list must be updated. For instance, a significant amount of literature references (too many) are reported dealing with landslide risk analysis, assessment and zoning and not enough references are reported dealing with risk analysis of tailing dams. Moreover, it is necessary to clearly state, before reporting the current landslide risk management practice, what its connection with the topic of the paper.

---

## Referee Comment (RC2) · Anonymous Referee #2 · 2 Oct 2018

**General comments**

The paper shows the results of a study focused on the management of risk (to people and buildings) posed by failures that might affect tailing dams. To this aim, a methodological approach – including six steps of activities – is proposed and applied to a case study in the Hubei Province (China). The addressed topic is relevant. However, much efforts should be done in carrying out a formal risk analysis and to properly evaluate the obtained results according to well-established risk tolerance criteria.

**Specific comments**

Section 2.2, The process of risk analysis – page 4, lines from 15 to 20. The reach (or run-out distance) is not an intensity parameter (related to the destructive power of the

mixture originated from the failure of a given tailing dam); it relates to the probability of the mixture reaching the elements at risk (Fell et al., 2005). Please, amend the text accordingly. Could the Authors better explain the meaning of the phrase: "the hazard of tailing dam failures are analysed according to the hazard classification of the debris flow"?

Section 2.2, The process of risk analysis – page 4, lines from 25 to 29. The equation (1) is incomplete; indeed, the risk (R) formula originally provided by Varnes (1984) is: $R = H \times V \times E$ where H is the hazard, E is the exposure (in the case of people) or the value (in the case of buildings) of a given element or a set of elements at risk whose vulnerability equals V. Please, modify accordingly.

Section 3.1.1, Parameters of moving tailings sand – page 5, lines from 13 to 21. This section is very poor. More details should be provided about the used physical model (for instance, in terms of values assigned to the mentioned ground slope and roughness; these values should be also justified) and numerical model. In particular, how does the Tsunami-Square model work? Is it based on the adoption of computational grid cells (useful for the estimation of the intensity parameter values)? What kind of rheological model has been adopted to simulate the behaviour of the mixture? Comparisons of data recorded in the experimental tests and in the numerical modelling should be synthesised in a Figure/Table in order to allow understanding how the values to be associated with relevant parameters have been retrieved.

Section 3.1.3, Hazard zoning in tailing dam failure – page 6, lines from 2 to 5. Only looking at Table 2 it is possible to understand how the intensity has been defined; this should be done in the manuscript. Anyway, the Authors should make explicit the criterion adopted to distinguish the three hazard levels (High, Moderate; Low). Furthermore, it is not clear if the so-called "mud depth" refers to the same computational time step in which the maximum velocity of the flowing mixture is recorded or represents the maximum value of depth attained by it (even in a different computational time step). In Table 2, the unit of measure of the (maximum) velocity should be provided.

Section 3.2, Vulnerability and Risk assessment of buildings – page 6, lines from 7 to 19. The Authors observe that "According to the actual situation, buildings are divided into four categories". Which? Reading the section, it seems that the identification of at risk areas precedes the vulnerability estimation. Furthermore, the vulnerability (of either people or buildings) is not defined; in particular, it is not explained if and how the vulnerability depends on the mixture intensity. A the same manner, the temporal-spatial probability of people at risk is posed equal to 0.6 without any clarification about its estimation (do the Authors refer to the average of persons at risk?). Finally, it is not clear how the equation (1) was used to calculate the risk and how the (four) risk levels have been established.

Section 4. Discussion: risk management schemes for extreme accidents – page 7, lines from 1 to 18. The Authors suggest the adoption of F-N curves to evaluate the acceptability/tolerability of calculated risk (to either people or buildings). In this regard, they claim that – in the case study at hand – "F is the probability of failure of a tailing dam". As a matter of fact, considering people at risk, F represents the cumulative probability (e.g. per year) that N or more lives will be lost; accordingly, F-N curves are usually adopted to evaluate the so-called "societal risk" (Fell et al., 2005; Leroi et al., 2005). From this point of view, it is not clear if the risk values obtained by the way of Eq. (1) just correspond to "societal risk" values. Moreover, the concept of reliability adopted by the Authors could fail if the correct definition of F is taken into account. Finally, the criterion used to individuate the F-N thresholds – useful to separate the F-N diagram in five zones (see Figure 6) – should be explained.

References

Fell R., Ho K.K.S., Lacasse S., Leroi E. 2005. Risk assessment and management. In O. Hungr, R. Fell, R. Couture, E. Eberhardt (eds.) Landslide Risk Management, pp. 3-26. London: Taylor and Francis.

Leroi E., Bonnard Ch., Fell R., McInnes R. 2005. A framework for landslide risk assessment and management. In O. Hungr, R. Fell, R. Couture, E. Eberhardt (eds.) Landslide Risk Management, pp. 159-198. London: Taylor and Francis.

Varnes D.J. 1984. Landslide hazard zonation: A review of principles and practice. The International Association of Engineering Geology Commission on Landslides and Other Mass Movements 1984. Natural Hazards, pp. 3-63. Paris (France):UNESCO.

Technical corrections

The reference "Corominas et al. (2013)" should be updated: Corominas J., van Westen C., Frattini P., Cascini L., Malet J.-P., Fotopoulou S., Catani F., Van Den Eeckhaut M., Mavrouli O., Agliardi F., Pitilakis K., Winter M.G., Pastor M., Ferlisi S., Tofani V., Hervàs J., Smith J.T. (2014). Recommendations for the quantitative analysis of landslide risk. Bulletin of Engineering Geology and the Environment, 73:209-263. doi:10.1007/s10064-013-0538-8

---

## Author Comment (AC1) · 12 Nov 2018

Thanks for the comments. That's very helpful. We have greatly revised and improved the article, added some missing parts and illustrated the diagrams and tables more precisely.

1, Comment "its content alternates back and forth between sections describing the features of the proposed procedure and sections illustrating the case study" Response: The article has already been restructured and separated the risk assessment steps and the case. We adjusted the order of parts of the content. The second and third parts mainly introduces the evaluation steps and the specific methods and operating procedures. The fourth part introduces the application of the specific case.

[Figure]

2, Comment "the role played by the different stakeholders does not clearly emerge from the . More details are needed on this central aspect of the proposed procedure." Response and changes in the manuscript: About the roles of different participants in the second part of the STEP 4, we introduce it and used to determine the upper and lower limits of the curve of tolerance risk and tolerable risk, as well as the supervision and supervision responsibility of each participant after the subsequent closure.

3, Comment "The originality of the overall procedure is not clear, as the key innovative elements proposed are not properly highlighted" and "clearly define if they have an essential role in the procedure or if they are only used to apply the procedure to the case study." Response and changes in the manuscript: The method of each step has been written in detail in the second part of each method below. The method described in process of each step is for the evaluation. The method in the third and fourth part is the method for the case.

4, Comment "Figure 2 should be restructured to better highlight the phases of the procedure described in the text." Response: The order of the left in Figure 2 is the general risk assessment process, the right is the extreme event of the risk assessment process, that is, in the case of failure to determine the failure probability of the risk analysis. It is used to distinguish between two different types of evaluation processes.

5, Comment "The results of the procedure applied to the case study must be described following the different phases of the procedure, as they are presented in Figure 2." Response and changes in the manuscript: After reconstruct the structure of the thesis, the third part and the fourth part of the paper are divided into two parts of risk evaluation and management, which is consistent with the order of the right evaluation six steps in Figure 2.

6, Comment "The role of the laboratory experiment must be better explained." Response and changes in the manuscript: In the third part of the risk assessment process, experimental groups and the different characteristics of each group are added.

More details about the experiment are given in the manuscript.

7, Comment "It is too long and not well balanced in terms of topics addressed." Response and changes in the manuscript: The introduction part simplifies lots of the content, strengthen the relationship with the theme of the article.
* * *

---

## Author Comment (AC2) · 12 Nov 2018

Thanks for the comments.That's very helpful.The order of the articles has been adjusted, the unreasonable places have been modified, and the missing parts and figures have been added.

1, Comment "The run-out distance is the strength parameter" Response and changes in the manuscript: this part has been modified.

2, Comment "the hazard of tailing dam failures are analyzed according to the hazard classification of the debris flow"? Response: Because the debris flow intensity proposed by Fiebiger is classified according to the depth of mud and relationship between the maximum flow rate and depth of mud. The debris flow geological hazard is used for

reference to risk zoning of tailings dam break-down, and the risk is divided according to its depth and velocity.

3, Comment "Section 2.2, The process of risk analysis – page 4, lines from 25 to 29. The equation(1) is incomplete" Response and changes in the manuscript: The equation (1) has been completed, which is $R = H \times V \times E$.

4, Comment "More details should be provided about the used physical model" Response: The values in the physical model and more details are added in article. A brief introduction to the principles of T-S method simulation is introduced. Changes in the manuscript: Based on the theory of solid and fluid mechanics, the Tsunami squares (T-S) method considers the volume and momentum conservation of the motion process of the flow (sliding) body, and establishes a theoretical model of the energy flow analysis of the flow (sliding) motion process. T-S method is to treat a mass of moving matter as a plane consisting of many small squares of the same size and with a certain thickness and velocity. Based on the continuity equation of Tsunami squares theory (volume conservation and momentum conservation equation), the appropriate calculation time step is chosen, the position, velocity, thickness and acceleration of each small squares movement at each time step is updated, and the motion characteristics of the material that the small squares simulate over time is derived.

5, Comment: "the Authors should make explicit the criterion adopted to distinguish the three hazard levels (High, Moderate; Low)." and "it is not clear if the so-called "mud depth" Response: the different level is distinguished by the product value of the maximum velocity and the mud depth. The mud depth is the depth of the final stop, velocity is the maximum velocity achieved during the movement.

6, Comments: "it seems that the identification of at risk areas precedes the vulnerability estimation. Furthermore, the vulnerability (of either people or buildings) is not defined" and the temporal spatial probability of people at risk is posed equal to 0.6 without any clarification about its estimation" Response and changes in the manuscript: The

vulnerability is modified in the article. Vulnerability needs to consider landslide intensity and susceptibility of elements at risk together. The index selection of susceptibility and its value are attached to the text. The susceptibility mainly considers the index of the risk assessment of the single landslide, that is, the structure of buildings, the maintenance of buildings and the service life of buildings. Mixture intensity mainly determines its hazard. The temporal spatial of people at risk (0.6) refers to the time in which people in the area live means the average of persons at risk. The hazard values are 1, 0.7 and 0.4 based on high, moderate and low level. Vulnerability is calculated based on model we have proposed in the article. The formula is listed in the article.

7, Comments: "As a matter of fact, considering people at risk, F represents the cumulative probability (e.g. per year) that N or more lives will be lost; accordingly, F-N curves are usually adopted to evaluate the so-called "societal risk"; "the Authors could fail if the correct definition of F is taken into account." Response and changes in the manuscript: In the original F-N curve, F represents the cumulative probability (e.g. per year) that N or more lives will be lost, at this time the F-N curve is for regional geological hazards, and the case in this article is only the single tailings dam failure, so F represents a single tailings dam failure probability.

8, Comments: "the criterion used to individuate the F-N thresholds – useful to separate the F-N diagram in five zones (see Figure 6) – should be explained." Response and changes in the manuscript: According the consequences of the disaster, and policies and regulations in the area, the standards of acceptable risk are usually determined by expert analysis. Residents need to be assured of the security at these sites, but this cannot be out of a range suggested by experts. Therefore, we propose that the upper (unacceptable) limit could be determined by experts, but that suggestions from residents should be considered for the lower (broadly acceptable) level. The government and mineral companies can act as intermediaries in determining the lower boundary of the acceptable risk. The risk buffer zone, for people and buildings, should be discussed by the multiparty representatives participating in the risk assessment.

| Model | Slope | Materials | Roughness | Burst shape |
|:-----:|:-----:|:---------:|:---------:|:-----------:|
| 1 | 10° | Plywood | Roughness | 1/4 |
| 2 | 10° | PVC | Smooth | 1/4 |
| 3 | 10° | Plywood | Roughness | 1/2 |
| 4 | 20° | Plywood | Roughness | 1/2 |

**Fig. 1.** Physical model experiment groups

| Hazard level | Mud depth (m) | Relationship | Mud depth(m) and maximum velocity(m/s) | Hazard assignment |
| --- | --- | --- | --- | --- |
| High | $H > 2.5$ m | OR | $VH \geq 2.5$ | 1 |
| Moderate | $0.5 \leq H < 2.5$ | AND | $0.5 \leq VH < 2.5$ | 0.7 |
| Low | $0.0 \leq H < 2.5$ | AND | $VH < 0.5$ | 0.4 |

**Fig. 2.** Scheme for dividing the geological environment into hazard zones after a tailings dam failure

| $S_{str}$ | Building structure | $S_{mai}$ | Building maintenance |
|---|---|---|---|
| 0.8 | Lightweight simple structure | 0.1 | Good |
| 0.6 | Brick-wood structure | 0.4 | Medium |
| 0.4 | Brick concrete structure | 0.7 | Bad |
| 0.2 | Reinforced concrete structure | 1 | Very bad |

| $S_{ser}$ | Ratio of service life and design life* |
|---|---|
| 0.05 | ≤0.1 |
| 0.1 | 0.1~0.4 |
| 0.3 | 0.4~0.6 |
| 0.5 | 0.6~0.8 |
| 0.7 | 0.8~1.0 |
| 0.8 | 1.0~1.2 |
| 1 | ＞1.2 |

**Fig. 3.** Susceptibility indexes value of structures

| Vulnerability | | Intensity | | |
|---|---|---|---|---|
| | | Low | Moderate | High |
| | | 0-0.4 | 0.4-0.7 | 0.7-1.0 |
| Low | 0-0.5 | 0-0.35 | 0.35-0.8 | 0.8-1 |
| Moderate | 0.5-0.65 | 0-0.5 | 0.5-0.85 | 0.85-1 |
| Moderate-High | 0.65-0.8 | 0-0.6 | 0.6-0.87 | 0.87-1 |
| High | 0.8-1.0 | 0-0.7 | 0.7-0.95 | 0.95-1 |

Susceptibility

*Green, yellow and red zone means the low, moderate,moderate-high and high level of vulnerability

**Fig. 4.** Vulnerability values of structures in dam failure